# High Multi-Environmental Mechanical Stability and Adhesive Transparent Ionic Conductive Hydrogels Used as Smart Wearable Devices

**DOI:** 10.3390/polym14235316

**Published:** 2022-12-05

**Authors:** Yuxuan Wu, Jing Liu, Zhen Chen, Yujie Chen, Wenzheng Chen, Hua Li, Hezhou Liu

**Affiliations:** State Key Laboratory of Metal Matrix Composites, School of Materials Science and Engineering, Shanghai Jiao Tong University, Shanghai 200240, China

**Keywords:** smart wearable devices, ionic conductive hydrogels, multi-environmental mechanical stability, multifunctional hydrogels

## Abstract

Ionic conductive hydrogels used as flexible wearable sensor devices have attracted considerable attention because of their easy preparation, biocompatibility, and macro/micro mechanosensitive properties. However, developing an integrated conductive hydrogel that combines high mechanical stability, strong adhesion, and excellent mechanosensitive properties to meet practical requirements remains a great challenge owing to the incompatibility of properties. Herein, we prepare a multifunctional ionic conductive hydrogel by introducing high-modulus bacterial cellulose (BC) to form the skeleton of double networks, which exhibit great mechanical properties in both tensile (83.4 kPa, 1235.9% strain) and compressive (207.2 kPa, 79.9% strain) stress–strain tests. Besides, the fabricated hydrogels containing high-concentration Ca^2+^ show excellent anti-freezing (high ionic conductivities of 1.92 and 0.36 S/m at room temperature and −35 °C, respectively) properties. Furthermore, the sensing mechanism based on the conductive units and applied voltage are investigated to the benefit of the practical applications of prepared hydrogels. Therefore, the designed and fabricated hydrogels provide a novel strategy and can serve as candidates in the fields of sensors, ionic skins, and soft robots.

## 1. Introduction

As a soft material composed of liquid and solid phases, hydrogel has excellent flexibility, biocompatibility, and sensing capability [1,2,3,4]. Over the last few decades, using hydrogel as smart wearable devices in the field of intelligent medical and health monitoring has attracted considerable attention [5,6,7,8]. In general, traditional conductive hydrogels always contain conductive fillers such as metal-based nanowires, conductive polymers, and carbon-based nanofillers [9,10,11]. However, those hydrogels are complex to prepare, and the incompatibility between gels and fillers severely limits their properties and practical applications.

Compared to traditional electronically conductive hydrogels, the conduction of ionic conductive hydrogels is mainly due to free-moving ions, which avoid the problems mentioned above [12,13,14,15]. In addition, they are usually transparent and have a similar conductive mechanism to that of the human tissues, giving them potential applications in the field of smart wearable devices such as sensory skins, human motion sensors, and personal healthcare diagnoses [16,17,18]. Zhang et al. [19] prepared ionic conductive hydrogels containing bacterial cellulose (BC), and the novel ion channels endow the hydrogels with stable and excellent sensing properties. Gupta et al. [20] fabricated silver-nanoparticle-loaded bacterial cellulose hydrogels for a potential wound-dressing application that can be directly attached to the skin surface. However, as a smart wearable material that can exhibit an effective strain-sensing and convert it into electrical signals, the ability of the hydrogels to deform with human skin in real time is essential. Tannic acid (TA) can be an efficient gelation binder for hydrogels since it has a molecular structure similar to that of mussels. For instance, Fan et al. [21] prepared dual-cross-linked single-network hydrogels with versatile adhesiveness by introducing TA as binders. Shao et al. [22] fabricated cellulose nanocomposite tough hydrogels with durable and repeatable adhesiveness and the authors ascribed it to the presence of catechol groups from the incorporated TA.

Besides, in some harsh environments, such as sub-zero and arid ones, hydrogels are usually damaged by dehydration or freezing, resulting in the loss of their functionality [23]. Therefore, it is also necessary to enhance the anti-freezing and moisturizing properties of hydrogels [24,25]. Lately, numerous studies focused on the preparation of hydrogels to resist freezing and moisturizing have been reported. He et al. [26] prepared conductive hydrogels that exhibited excellent anti-freezing and moisturizing properties by introducing a water–glycerol dispersion medium. Morelle et al. [27] reported anti-freezing hydrogels with tough polyacrylamide–alginate double networks containing Ca^2+^ ions.

Nevertheless, among the studies of transparent conductive hydrogels that have been reported, most of them paid tremendous attention to the response and recognition of various mechanical signals, as well as to the exploration of multifunctions. Besides, those studies only concentrated on the stable conductivity, high sensitivity, and microstructure of ionic conductive hydrogels, without focusing on the specific factors related to their sensing performance and the mechanism of strain sensing. Hence, it is urgently needed to design and prepare transparent multifunctional/multi-response conductive hydrogels and deeply investigate the factors affecting their sensing performance to meet practical requirements.

Herein, we prepared a new hydrogel with high stretchability, excellent adhesion, and stable mechanical sensing properties in freezing and arid environments, which consisted of TA@BC, acrylic acid (AA), and 2-acrylamide-2-methylpropane sulfonic acid (AMPS). TA-coated BC improved its uniform distribution in the hydrogel, and natural polyphenol TA formed hydrogen bonds with polymer that also toughen the polymer network [28]. Moreover, a large number of catechol groups supplied by TA can bind to both inorganic and organic surfaces through the formation of reversible noncovalent or irreversible covalent interactions [21,29,30]. The ionic conductive hydrogels can adapt to different deformations (including stretching and compression) and present a stable response to various stimuli through relative resistance changes (RRCs) and the hydrogel can maintain high ionic conductivities of 0.36 S/m at −35 °C. When applicated as a biosensor, the prepared hydrogel not only exhibits an excellent human-motion detection function but also has a sensitive response to micro-deformations such as pulse and writing. Additionally, to promote practical applications of hydrogels, we analyzed the sensing performance of the hydrogels under different tensile strains and bending strains, and explored the effects of two main factors (metal ion concentration and applied voltage). In summary, the prepared hydrogels exhibited excellent mechanical properties, great sensing performance, and strong adhesion, providing a novel strategy for the development of next-generation smart materials, including ionic skin, health monitoring, soft robots, etc. The exploration also provides a new prospect for the practical applications of ionic conductive hydrogels.

## 2. Experimental Section

### 2.1. Synthesis of the Hydrogel and Materials

The concentration of 1 wt% BC suspension was first diluted to the desired concentration (1 wt%, 0.5 wt%, 0.25 wt%) and stirred to disperse. Then TA was added to the BC solution at a mass ratio of BC: TA = 1:2 and stirred magnetically for 6 h at room temperature to finally obtain the TA@BC suspension. Then, 0.5 g AA and 0.5 g AMPS were added to 4 mL of the above-prepared TA@BC suspension and stirred for 0.5 h. Subsequently, *N, N*-methylene bisacrylamide (BIS), CaCl_2_, and AMPS were added to the homogeneous solution and stirred for 0.5 h. The mixed solution was degassed by vacuum and stirred again under a N_2_ atmosphere for 0.5 h. Finally, the hydrogel was fabricated through radical polymerization under UV irradiation (365 nm, 20 W) for 0.5 h using a xenon light source (Shanghai Yuming Instrument Co., Ltd., Shanghai, China). AA was obtained from Sinopharm Chemical Reagent (Shanghai, China). BIS and UV initiator were purchased from Shanghai Macklin Biochemical Co., Ltd. (Shanghai, China). In addition, AMPS was purchased from Bide Pharmatech, Ltd. (Shanghai, China), and TA and CaCl_2_ were purchased from Aladdin Reagent Co., Ltd. (Shanghai, China). BC was obtained from EneRol Nanotechnologies Co., Ltd. (Ningbo, China). 

### 2.2. Characterization

The infrared spectra were obtained by scanning the samples in the wave number range of 450–4000 cm^−1^ using an FT-IR spectrometer (Spectrum 100, PerkinElmer Co., Ltd., Waltham, MA, USA). The microstructures and EDS maps of the freeze-dried hydrogels and TA@BC fibers were observed using field-emission SEM (Mira3, TESCAN Co., Ltd., Brno, Czech Republic). Finally, the surface morphology images of the fibers and the thickness distribution profiles were recorded using AFM (MFP-3D, Oxford Instruments Co., Ltd., Oxford, UK).

### 2.3. Mechanical Test

The mechanical tests were conducted at ambient and sub-zero temperatures, using a microcomputer-controlled electronic universal testing machine (LD23 503, Shenzhen, China). The tensile sample was a strip (5 mm × 30 mm × 1 mm, width × length × thickness) and the test speed was 20 mm/min. The compression sample was a cylinder (diameter: 20 mm, height: 10 mm) and the test speed was 10 mm/min.

### 2.4. Anti-Freezing Property and Moisturizing Property Tests

During the test of freezing and ice-melting temperatures, the hydrogel was first cooled from 20 °C to −70 °C at −5 °C/min and subsequently heated to 20 °C. The experimental data was obtained by DSC (DSC 2500, TA Instruments Co., Ltd., Newcastle, DE, USA).

As for the moisturizing properties of this hydrogel, the weight change was recorded at room temperature and 35–40 RH% over 120 h.

### 2.5. Adhesion Performance Test

The shear adhesion strength of the gels was tested using a microcomputer-controlled electronic universal testing machine (LD23 503, Shenzhen, China). The gel samples with the shape of 10 mm × 10 mm × 2 mm (width × length × thickness, respectively) were tightly adhered to various material substrates, such as PET plastic, glass, wood, PTFE sheet, and iron sheet, and the shear adhesion strength was tested at a tensile speed of 10 mm/min. The adhesion energy (W_adh_) of the gel is calculated as follows: W_adh_ = $\int Fdx/A_{max}$, where *F* is the shear force, *x* represents the displacement and *A_max_* is the maximum contact area of the gel.

### 2.6. Electrical Properties Testing

We used a precision semiconductor parameter analyzer (Agilent 4156C, Keysight Technologies Co., Ltd., Santa Rosa, CA, USA) to monitor the real-time electrical signal changes of the hydrogel sensor. The RRCs of the sample were calculated as follows: ∆R/R_0_ (%) = (R − R_0_)/R_0_ (%), where R_0_ and R are the original and real-time resistance of the prepared gel sample during the measurement procedures. The GF is defined as GF = (R − R_0_)/R_0_)/ε, where ε is the strain. The conductivity changes of the hydrogels were monitored in the temperature range of −40−20 °C using an integrated physical property measurement system (PPMS-9T, Alfa Chemistry Co., Ltd., New York, NY, USA) in variable temperature mode.

## 3. Results and Discussion

### 3.1. Design and Fabrication of Hydrogel

With good biocompatibility, high sensitivity, and stable ionic conductivity, ionic conductive hydrogels are considered as potential candidates for smart wearable devices, such as sensory skins, human motion sensors, and personal healthcare diagnoses. However, poor mechanical properties limit their applications. To solve this problem, in this study, BC and P(AA-AMPS) polymer chains with excellent biocompatibility were used to form a double network hydrogel, and the hydrogel with physical and chemical cross-linking can be easily prepared using a one-pot method (Figure 1a). Briefly, AA, AMPS, *N, N*-methylene bisacrylamide (BIS), and CaCl_2_ were added to the prepared TA@BC suspension. Under UV irradiation (365 nm, 20 W) for 0.5 h, the hydrogel was then fabricated through radical polymerization, where the name of the gel is defined as P(AA-AMPS)-TA@BA_x_-Ca^2+^ and x represents the solid content of TA@BC. The specific components of the hydrogel are shown in Appendix A.

In this work, BC and P(AA-AMPS) polymer chains formed a double network structure to enhance the mechanical properties of gels. TA not only acted as adhesion units to endow the material with good adhesion properties but also coated the BC surface to improve the dispersion [22]. The high concentration of Ca^2+^ ions enhanced the hydrogels’ strength through the formation of metal-carboxyl coordination bonds and also endowed the hydrogels with stable ionic conductivity. Moreover, the hydration between the Ca^2+^ ions and water could enhance the moisture retention and freezing resistance of the gels [31,32].

In the AFM and SEM images (Figure 1b,c), the fiber of the TA@BC surface became rougher and appeared to have a smaller aspect ratio after TA coating [22], and the thickness of the TA@BC reached 31.9 nm in the thickness curve, which was 2.5 times that of the uncoated BC (12.9 nm). Besides, a large number of slightly smaller and uniform porous structures could be observed in P(AA-AMPS) after freeze drying (Figure 1b). After adding TA@BC, the double network structure was formed by TA@BC and P(AA-AMPS), which enlarged the pores inside of hydrogel. Meanwhile, flocculent fibers can be clearly observed in the pores. However, after adding a high concentration of Ca^2+^, there is only a small and sparse porous structure on the surface of P(AA-AMPS)-TA@BC-Ca^2+^. The reason was that the high concentration of Ca^2+^ ions reduced the freezing point of water in the hydrogel during the freezing process since it formed smaller ice crystals. The uniform distribution of S and N elements in the prepared hydrogels can be observed in EDS (Appendix A), which showed the uniform distribution of P(AA-AMPS) polymer chains in the hydrogel with excellent physical properties. In addition, the uniform distribution of the Ca element indicated that Ca^2+^ ions could easily pass through the porous structure of the hydrogels, ensuring the stable conductivity of the gels.

The Fourier transform infrared spectrometer (FT-IR) spectra of AA, AMPS, BC, TA, TA@BC, P(AA-AMPS), P(AA-AMPS)-TA@BC, and P(AA-AMPS)-TA@BC-Ca^2+^ are presented in Figure 2. Compared with unmodified BC fiber, the spectrum of TA@BC showed some remarkable characteristic peaks of TA, such as C=O(1716 cm^−1^), C=C(1606 cm^−1^, 1535 cm^−1^, 1448 cm^−1^), and O-H(1322 cm^−1^, 1203 cm^−1^) bonds, [22,33] which represented the carboxyl group, benzene ring, and phenolic hydroxyl group in TA, respectively. In addition, after TA is coated on the surface of BC, the absorption peaks of -OH in TA (3425 cm^−1^) and BC (3349 cm^−1^) shifted to 3114 cm^−1^ of TA@BC, indicating that a strong hydrogen bond force was formed between TA and BC. Moreover, the changes of TA@BC in the AFM and SEM images (Figure 1b,c) and the thickness distribution curve (Figure 1d) confirmed that TA was successfully coated on the surface of the BC.

For the hydrogel of P(AA-AMPS), the absorption peaks at 2976 and 1452 cm^−1^ represented the O-H [34] and C–N [35] bonds of the AA and AMPS, respectively, which showed the successful polymerization of the P(AA-AMPS) hydrogel, and the C=O at 1704 cm^−1^ was significantly enhanced when the fiber of TA@BC was added into the P(AA-AMPS) hydrogels. After adding Ca^2+^, the absorption peak of C=O shifted from 1704 cm^−1^ to 1700 cm^−1^, demonstrating the form of metal-carboxyl coordination bonds between Ca^2+^ ions and the carboxyl group of AA and AMPS [31,34]. Besides, the absorption peaks at 3297 cm^−1^ in the P(AA-AMPS)-TA@BC and P(AA-AMPS)-TA@BC-Ca^2+^ hydrogels were associated with the hydrogen bonds between TA@BC and P(AA-AMPS), respectively.

### 3.2. Mechanical Properties of Hydrogel

Owing to the uniform intermolecular and intramolecular interactions in the P(AA-AMPS)-TA@BC-Ca^2+^ gel and the double network structure formed by BC and P(AA-AMPS), the hydrogel performed excellent mechanical properties. As a fiber with a large aspect ratio, BC showed high elastic modulus and strength [19], and its content has a significant effect on the strength of the hydrogel. With the increase in TA@BC content from 0.25 wt% to 0.5 wt%, the tensile strength of the P(AA-AMPS)-TA@BC-Ca^2+^ hydrogel increased from 24.1 kPa to 83.4 kPa, and the elongation at break of the hydrogel increased from 756% to 1236% (Figure 3a). However, when the content of TA@BC increased to 1 wt%, the strength and strain of the hydrogel significantly decreased to 60.1 kPa and 1000%, respectively, which might be attributed to the large amount of BC and TA reducing the transparency of the hydrogel and hindering the UV-induced radical polymerization of AA and AMPS. Thus, the mechanical properties of the gel were reduced. The P(AA-AMPS)-TA@BC_0.5_-Ca^2+^ gel was subjected to multiple cyclic tensile tests, and the results showed the stress–strain curves of the gel in the cycle (600% strain, 5 cycles) basically overlapped and had very small hysteresis, which indicated the good elasticity of the hydrogel. When TA@BC and Ca^2+^ were gradually added into the P(AA-AMPS) system, the maximum stress and strain of the hydrogel were simultaneously increased because it was affected by the double network structure (formed by TA@BC fibers and polymer network) and the dynamic coordination bonds (formed by Ca^2+^ and carboxyl groups). Moreover, the tensile strength and maximum elongation of P(AA-AMPS)-TA@BC_0.5_-Ca^2+^ were 6.7 and 3.4 times higher than that of P(AA-AMPS) hydrogel, respectively (Figure 3c). In addition, the tensile properties of P(AA-AMPS)-TA@BC-Ca^2+^ hydrogels with different Ca^2+^ concentrations were tested. Figure 3d shows that the hydrogels with a Ca^2+^ concentration of 3M exhibited the best mechanical properties (stress and strain), which can be attributed to the more metal-carboxyl coordination bonds in the hydrogel, and the toughness of the P(AA-AMPS)-TA@BC-Ca^2+^ hydrogels with different concentrations of TA@BC is shown in Appendix A.

The prepared hydrogels also showed excellent compression properties. Here, the compression properties of hydrogels with different TA@BC contents were tested, and the compression process is shown in Appendix A. As the TA@BC content increased to 0.5 wt%, the strength of the gel increased from 105 kPa to 207 kPa, and the gel with 0.5 wt% content was twice as strong as the pure gel at 80% compressive strain (Appendix A). However, when the content of TA@BC increased to 1 wt%, the compressive strength of the hydrogel dropped to 153 kPa, which was consistent with the tensile properties of the gel. The continuous compression-release performance of the hydrogel was tested by 80% strain, 5 cycles (Appendix A). During compression-release cycles, the stress–strain curve and hysteresis loop of the hydrogel completely coincide. Therefore, the prepared hydrogel exhibited sufficient toughness to withstand an 80% continuous cyclic compressive deformation. When the external force was removed, the hydrogel could recover to its original state, indicating that it had good elasticity and fatigue resistance. Then, the compression properties of P(AA-AMPS), P(AA-AMMPS)-TA@BC_0.5_, and P(AA-AMPS)-TA@BC_0.5_-Ca^2+^ were all tested (Appendix A). Similar to the results for tensile properties, the addition of TA@BC and Ca^2+^ significantly increased the compressive properties of the hydrogel. In view of the fact that the hydrogel system showed the best excellent mechanical properties when the content of TA@BC was 0.5 wt%, the P(AA-AMPS)-TA@BC_0.5_-Ca^2+^ gel material was preferred as the main research object in the following studies.

### 3.3. Mechanical and Conductive Stability of the Hydrogel in Freezing and Arid Environments

As a flexible material containing a large amount of water, hydrogels are prone to freezing or dehydration in cold or dry environments, resulting in the loss of their functionality. For example, in a cold environment, the hydrogel will be brittle because of the crystallization of water inside the gel, which then affects the mechanical properties and the ionic conductivity of the ionic conductive hydrogel. Therefore, maintaining the properties of the materials in multiple environments is a challenge for the development of flexible hydrogels. One of the ways to effectively resist freezing and moisturizing is to introduce anti-freezing components, such as high-concentration Ca^2+^/Li^+^, ethylene glycol, dimethyl sulfoxide [36,37,38], etc. In this work, the addition of 3 mol/L Ca^2+^ endowed the hydrogel with excellent anti-freezing properties. To measure the freezing resistance of the material, the freezing points of P(AA-AMPS)-TA@BC_0.5_-Ca^2+^ hydrogels with different Ca^2+^ concentrations were recorded by DSC during heating from −50 °C to 20 °C. We define the peak of the ice crystals dissolving and absorbing heat as the freezing point of the gel. During the heating process of the DSC curve, the pure gel P(AA-AMPS) showed a large endothermic peak at 0 °C (freezing point of P(AA-AMPS) hydrogel), which was owing to the melting of the ice crystals in the sample. However, when Ca^2+^ ions were added to the gel, the freezing point of the P(AA-AMPS)-TA@BC_0.5_-Ca^2+^ hydrogel decreased significantly. As the concentration of Ca^2+^ ions in the hydrogel increased from 1M to 3M, the freezing point of the P(AA-AMPS)-TA@BC_0.5_-Ca^2+^ gel decreased from −11 °C to −33 °C (Figure 4a), showing a significant anti-freezing effect. The lower freezing point of the hydrogel was due to the fact that the high concentration of the salt solution will increase the osmotic pressure and reduce the freezing point based on the colligative properties of the solution. Additionally, the hydration of Ca^2+^ will affect the hydrogen bonds of water molecules. These factors all affect the formation of ice crystals inside the gel (Figure 4a) [33,39]. As shown in Figure 4b, the P(AA-AMPS) and P(AA-AMPS)-TA@BC_0.5_-Ca^2+^ gels were simultaneously frozen at −20 °C for 24 h. The moisture inside the pure P(AA-AMPS) gel crystallized during the cold processes, and the gel changed from transparent to white. Meanwhile, the P(AA-AMPS) gel became fragile and easily broke during the stretching process. By contrast, the P(AA-AMPS)-TA@BC_0.5_-Ca^2+^ gel remained transparent and flexible after freezing, and could be easily twisted and stretched. To clearly characterize the mechanical stability of the hydrogel at low temperatures, we tested the tensile properties of the hydrogel in the frozen state and normal state (Figure 4c). The results showed that the fracture stress and strain of the gel stored at −20 °C for 24 h can still reach 68.2 kPa and 1013.8%, which are 81.7% and 82.0% of the normal state, respectively. The performance proves that the hydrogel material has excellent anti-freezing performance and stability.

Besides, the P(AA-AMPS)-TA@BC_0.5_-Ca^2+^ hydrogel exhibited not only excellent anti-freezing properties but also sufficient moisturizing properties. Within 120 h in an open environment at room temperature, the hydrogel P(AA-AMPS) retained only 14.3% of its original weight, showing obvious shrinkage and dehydration (Figure 4e). However, the P(AA-AMPS)-TA@BC_0.5_-Ca^2+^ gel still retained 70% of its initial weight, and is less deformed compared to the P(AA-AMPS) gel (Figure 4d). Meanwhile, after being placed in the open environment for 120 h, the elongation of the prepared hydrogel was still 1025.3%, and the stress was 75.2 kPa, which were 83% and 91% of their initial state, respectively (Figure 4f). The excellent moisture-retention performance of the P(AA-AMPS)-TA@BC_0.5_-Ca^2+^ hydrogel was due to the fact that the addition of CaCl_2_ could reduce the vapor pressure of water and effectively inhibit the evaporation of water in the hydrogel. Specifically, the hydration between Ca^2+^/Cl^−^ and water molecules effectively inhibited the evaporation of water molecules. Therefore, the P(AA-AMPS)-TA@BC_0.5_-Ca^2+^ hydrogel exhibited good moisturizing properties (Appendix A).

The high concentration of Ca^2+^ ions in the hydrogel not only endowed the material with great anti-freezing and moisturizing properties but also with excellent ionic conductivity. The ionic conductivity of the P(AA-AMPS) and P(AA-AMPS)-TA@BC_0.5_-Ca^2+^ hydrogels at different temperatures was tested by a comprehensive physical property measurement system (Appendix A). Compared with the P(AA-AMPS) gel, the ionic conductivity of the P(AA-AMPS)-TA@BC_0.5_-Ca^2+^ hydrogel reached 1.92 S/m at room temperature, which was 80 times higher than that of the P(AA-AMPS) gel (0.024 S/m), and the conductivity of the P(AA-AMPS) gel dropped by an order of magnitude (from 0.024 S/m to 0.001 S/m) as the temperature decreased from 20 °C to −35 °C. Since the freezing point of the P(AA-AMPS) gel is 0 °C, while the ion migration rate decreased with the change of temperature, the formation of ice crystals also hindered the migration of ions, thereby reducing the conductivity of the hydrogel. By contrast, the P(AA-AMPS)-TA@BC_0.5_-Ca^2+^ hydrogels still exhibited an ionic conductivity of 0.36 S/m at −35 °C, which was 360 times higher than that of the P(AA-AMPS) gel. The high conductivity indicated that the material showed great anti-freezing properties and excellent ionic conductivity above the freezing point.

### 3.4. Adhesion Properties of Hydrogel

As a wearable device in direct contact with the human body, it is highly desirable to improve the adhesion of the hydrogel to human skin or a prosthesis. Strong and repeatable adhesion is conducive to the conformal attachment of the strain sensor and avoids the occurrence of interface delamination under large deformation to ensure the accuracy of monitoring. The presence of catechol in TA is believed to fulfill the dual role of interfacial binding and the solidification of the adhesive proteins [40]. Catechol is capable of diverse chemistries, which enable it to bind to both organic and inorganic surfaces through the formation of reversible non-covalent or irreversible covalent interactions [29,30]. Based on the catechol groups of oxidized polyphenols mimicking the mussel adhesion mechanism, the TA@BC hydrogels exhibited unique adhesiveness to a wide range of surfaces [22]. The P(AA-AMPS)-TA@BC_0.5_-Ca^2+^ hydrogels showed robust adhesion to various substrates such as plastic, glass, PTFE, ceramic, rubber, and iron (Figure 5a). These results indicated that the hydrogel could be adhered directly on prosthetic or robotic surfaces made of various materials without any other additional processing. Moreover, we can evaluate the adhesion properties of hydrogels by measuring the tensile adhesion strength between the hydrogel and the matrix. Among the matrix materials, the P(AA-AMPS)-TA@BC_0.5_-Ca^2+^ hydrogel showed the highest adhesion strength and adhesion force to wood, reaching 27 kPa and 204 N/m, respectively (Figure 5c,d). The main adhesion units of the hydrogel were TA containing abundant of o-phenolic hydroxyl groups, which can be tightly combined with the surface of the material through a large number of hydrogen bonds, especially a matrix with a large number of hydroxyl groups such as wood and glass [22,33]. More importantly, dynamic hydrogen-bonding interactions endowed TA@BC hydrogels with biocompatibility and repeatable adhesion without sacrificing other desirable properties, which were attractive for practical applications as wearable strain sensors.

### 3.5. Electrical Sensing Properties of Hydrogel

Owing to the dual network structure and uniform conductive ions, the hydrogel not only exhibited excellent mechanical properties but could also serve as a sensing medium to transmit the relative resistance changes (RRCs) caused by the change of the conductive pathway. The conductivity of the P(AA-AMPS)-TA@BC_0.5_-Ca^2+^ hydrogel reached 1.92 S·m^−1^ because of the existence of a large number of free Ca^2+^ ions (Appendix A).

To visually display the sensing ability of the P(AA-AMPS)-TA@BC_0.5_-Ca^2+^ hydrogel, we connect the gel strip in the closed circuit with a LED lamp under a direct current voltage of 4.5 V (Figure 6a). The brightness of the LED lamp was gradually decreased when the hydrogel was stretched, which indicated the increase in resistance. To show the real-time mechanical response properties of the hydrogels more clearly, the RRC vibration of the hydrogels was measured under diverse degrees of tensile strains. As shown in Figure 6b,c, both in small and large strains, the RRCs of the hydrogel changed obviously without delay. For a single stretching and releasing process, the RRCs value drops to the original state without obvious hysteresis after removing the tensile load.

To verify the effect of different concentrations of Ca^2+^ ions on the sensing performance of hydrogels, the RRCs of P(AA-AMPS)-TA@BC_0.5_-Ca^2+^ with 1M, 2M, and 3M Ca^2+^ concentrations under diverse degrees of tensile strains are shown in Figure 6d–f. When the ion concentration increased from 1 M to 3 M, the conductivity of the gel increased from 0.42 to 1.92 S/m (Appendix A). However, at the same strain, hydrogels with higher conductivity had no significant effect on the corresponding change in RRC values. The reason was that the ion concentration in the hydrogel did not change the number of charges during stretching. Therefore, as for the same ions, higher conductivity was not a relevant factor for the sensitivity of the gel sensor. Whereas, when the voltage gradually increased from 0.5 V to 1 V, the RRC value at the same strain increased, and the gauge factor (GF) of the hydrogel at 500% strain increased from 1.49 to 2.18. Due to this, the initial current of the gel increased as the voltage increased, resulting in an expansion of the current variation range. Additionally, the migration rate of Ca^2+^ ions in the hydrogel was affected by the voltage, thereby enhancing the sensitivity of the hydrogel to the strain responses (Figure 6g–i). Hence, as for ionic conductive hydrogels, within a specific range, the migration rate of ions is the main factor related to the sensing performance of the hydrogel.

The strain-sensing ability of 3M P(AA-AMPS)-TA@BC_0.5_-Ca^2+^ hydrogel was calculated using the value of GF. When the tensile strain changed from 2.5% to 900%, the GF value of the gel increased from 0.91 to 2.91 (Appendix A), which indicated that in the range of 2.5% to 900%, the P(AA-AMPS)-TA@BC_0.5_-Ca^2+^ hydrogel exhibited reliable, stable and sensitive strain-sensing performance.

In addition, the hydrogel also presented a sensitive and stable response to the stretching frequency from 0.16 Hz to 1 Hz (Appendix A), and under 400% strain, the sensing response time and recovery time of the hydrogel were 460 and 500 ms, respectively, which satisfied the requirements of human motion detection (Appendix A). For the long-term use of hydrogel sensors, the ability to stably transmit real-time electrical signals during repeated stretch–release cycles is critical. As for the P(AA-AMPS)-TA@BC_0.5_-Ca^2+^ hydrogels, stable mechanical response properties could be maintained under 400% strain after 300 stretch–release cycles (Appendix A), and during the test, there was no obvious shift in the RRCs of the hydrogel, indicating that the P(AA-AMPS)-TA@BC_0.5_-Ca^2+^ hydrogel had excellent mechanical response properties and long-term reusability.

### 3.6. Behavioral Monitoring of Human Movement and Physiological Signals by Hydrogel

With high mechanical stability, strong adhesion, great ionic conductivity, and excellent sensitivity, the P(AA-AMPS)-TA@BC_0.5_-Ca^2+^ hydrogel is a potential candidate for human motion detection and physiological signal monitoring.

As shown in Figure 7c, the adhered P(AA-AMPS)-TA@BC_0.5_-Ca^2+^ hydrogel on the finger joint rapidly responded to each bending (0° to 90°) angle by RRC values. Additionally, the stable current signal from the ionic conductive hydrogel could monitor the process of instantaneous change of different bending angles. When the bending angle of the finger abruptly varied from 30° to 60°, the RRC value of the gel increased from 0.968 to 1.912 without delay, and when it changed from 90° to 0°, the RRC value quickly recovered to the original state, which indicated that the hydrogel could maintain stable and highly sensitive response characteristics during continuous motion detection (Figure 7a). In Figure 7b, the finger’s slow and continuous bending process was monitored. After fitting, the results showed an obvious linear function between the output RRC value and the angle (R^2^ = 0.9959). Hence, the P(AA-AMPS)-TA@BC_0.5_-Ca^2+^ hydrogel can be considered as an ideal candidate for detecting human motion.

In addition to macroscopic deformations related to human motion detection, the P(AA-AMPS)-TA@BC_0.5_-Ca^2+^ hydrogel could also respond to various physiological signals. When writing on the surface of the hydrogel, the specific characteristics of the handwriting, like writing order and writing strength, can be translated into different electrical signals, which means that the hydrogel can distinguish different handwritten letters. As shown in Figure 7e, we connected the copper electrodes on the left and right sides of the hydrogel, and then covered the upper and lower sides with PET film to prevent water loss and scratched the gel. Three numbers were written on the hydrogel surface, and the corresponding RRC signals differed and had their own characteristics (Figure 7f). Moreover, the hydrogel could also be used to monitor subtle changes of the human pulse. In Figure 7g, the P(AA-AMPS)-TA@BC_0.5_-Ca^2+^ hydrogel showed the ability to clearly identify the pulse before and after exercise. We took the ionic conductive hydrogel adhered to the wrist to monitor the subtle changes of the volunteers’ pulse before and after running for 5 min. We found that the corresponding RRC value increased significantly after running, indicating that the pulse beat more vigorously after exercise. The experiment also proved the high sensitivity of the gel for micro-motion detection. Therefore, the P(AA-AMPS)-TA@BC_0.5_-Ca^2+^ hydrogel can be used to detect the physiological signals of the human body, and return real-time electric signals in the form of RRC values.

## 4. Conclusions

In summary, a facile strategy was employed to prepare the multiple-stimuli-responsive hydrogels, specifically employing acrylic acid (AA), 2-acrylamide-2-methylpropane sulfonic acid (AMPS), and high-modulus bacterial cellulose (BC) as a skeleton of a double network, and introducing tannic acid (TA) as adhesion units and anhydrous calcium chloride (CaCl_2_) as anti-freezing components. Under the UV initiation, a dual-network ionic conductive hydrogel with excellent adhesion, anti-freezing, and moisturizing functions and high stretchability was prepared by radical polymerization. The hydrogel showed excellent mechanical properties (83.4 kPa and 1236%) because of the metal-carboxyl coordination bonds and the double network structure. Moreover, by introducing the high concentration of Ca^2+^ ions, the hydrogel exhibited great anti-freezing performance from −33 °C to 0 °C, excellent moisture-retention performance at room temperature (70 wt%, 120 h), and stable ionic conductivity (1.92 S/m). In addition, the hydrogel can be tightly adhered to the surface of the human body because of the unique structure of TA, and can sensitively detect human motion, physiological signals, and the strains in the range of 2.5–900%. Interestingly, the RRC signals of the hydrogel showed a linear correlation with the tensile strain and bending angle during both continuous stretching and bending. Besides, the experiment on the sensing performance under different concentrations of Ca^2+^ and different voltages further demonstrated that the strain-sensing ability of ionic conductive hydrogels was mainly related to the ion migration rate, rather than the conductivity or ion concentration. To summarize, the hydrogel exhibits robust and repeatable adhesion, excellent anti-freezing and moisturizing properties, and stable mechanical sensing properties, which are expected to be applied in the fields of a new generation of electronic-skin soft robots and human motion detection.

## Figures and Tables

**Figure 1 polymers-14-05316-f001:**
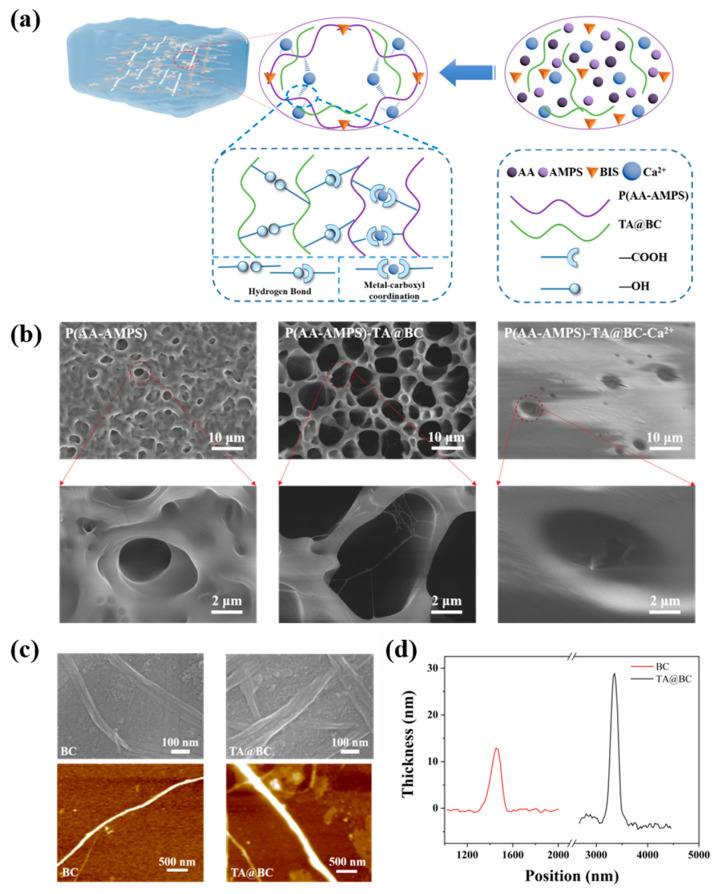
(**a**) Schematic diagram of hydrogel structure. (**b**) SEM images of P(AA-AMPS), P(AA-AMPS)-TA@BC, P(AA-AMPS)-TA@BC-Ca^2+^ hydrogel. (**c**) AFM images of BC and TA@BC cellulose. (**d**) Thickness distribution curve of BC and TA@BC in AFM.

**Figure 2 polymers-14-05316-f002:**
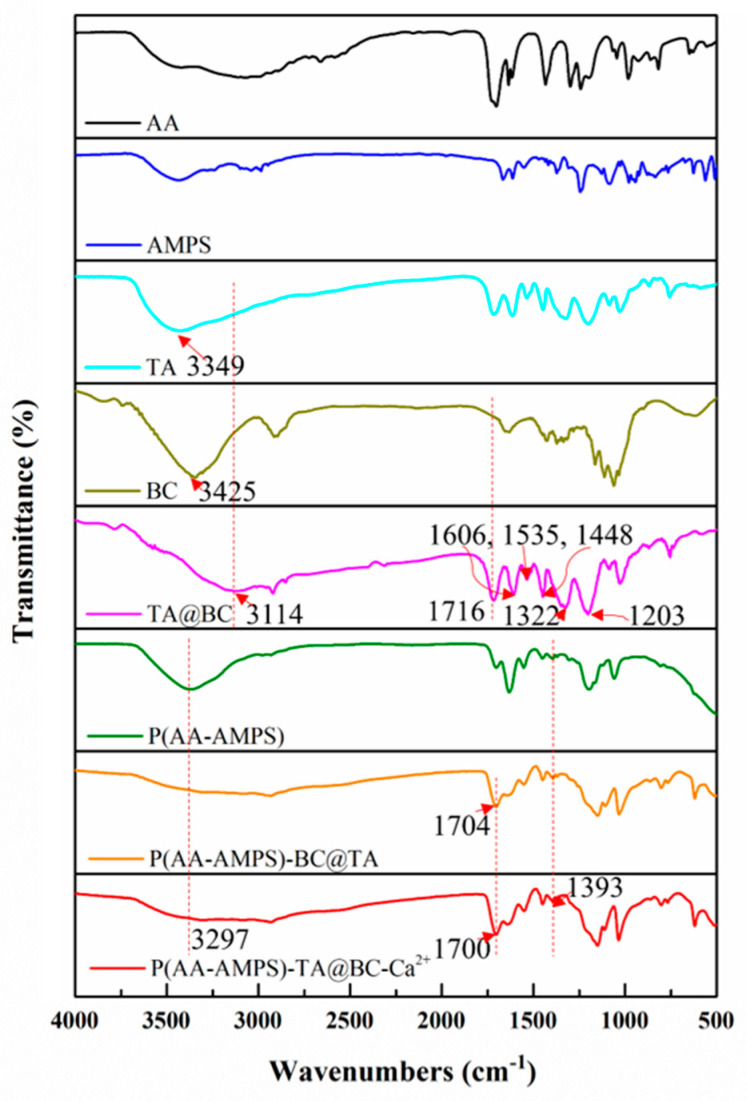
The FT-IR spectra of AA, AMPS, TA, BC, TA@BC, P(AA-AMPS), P(AA-AMPS)-TA@BC, and P(AA-AMPS)-TA@BC.

**Figure 3 polymers-14-05316-f003:**
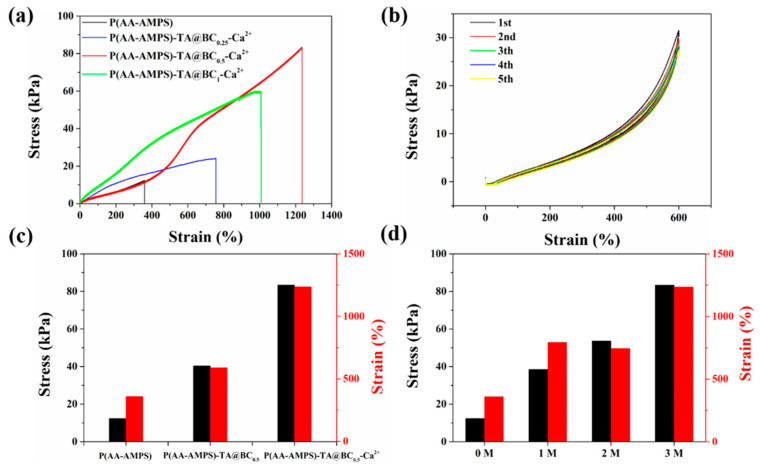
Tensile properties of hydrogel. (**a**) Tensile stress–strain curves of P(AA-AMPS)-TA@BC-Ca^2+^ hydrogels with different concentrations of TA@BC; (**b**) stretch–release cycle (600%, 5 cycles) of P(AA-AMPS)-TA@BC_0.5_-Ca^2+^ hydrogel; (**c**) tensile stress–strain histogram of P(AA-AMPS), P(AA-AMPS)-TA@BC_0.5_, and P(AA-AMPS)-TA@BC_0.5_-Ca^2+^ hydrogels (black columns: stress, red columns: strain); (**d**) tensile stress–strain histogram of P(AA-AMPs)-TA@BC-Ca^2+^ hydrogels with different concentrations of Ca^2+^ ions (black columns: stress, red columns: strain).

**Figure 4 polymers-14-05316-f004:**
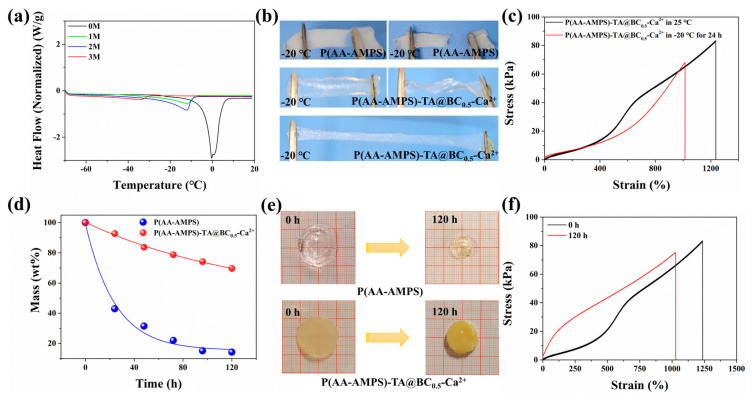
Anti-freezing and moisturizing properties of the hydrogel. (**a**) DSC curves of P(AA-AMPS)-TA@BC_0.5_-Ca^2+^ hydrogels with different Ca^2+^ concentrations during the heating process; (**b**) optical images of P(AA-AMPS) and P(AA-AMPS)-TA@BC_0.5_-Ca^2+^ hydrogel at −20 °C; (**c**) stress–strain curves of P(AA-AMPS)-TA@BC_0.5_-Ca^2+^ hydrogel before and after being stored at −20 °C for 24 h; (**d**) mass curves of P(AA-AMPS) and P(AA-AMPS)-TA@BC_0.5_-Ca^2+^ hydrogels at room temperature; (**e**) optical images of P(AA-AMPS) and P(AA-AMPS)-TA@BC_0.5_-Ca^2+^ hydrogel in room temperature; (**f**) stress–strain curves before and after 120 h storage in room temperature.

**Figure 5 polymers-14-05316-f005:**
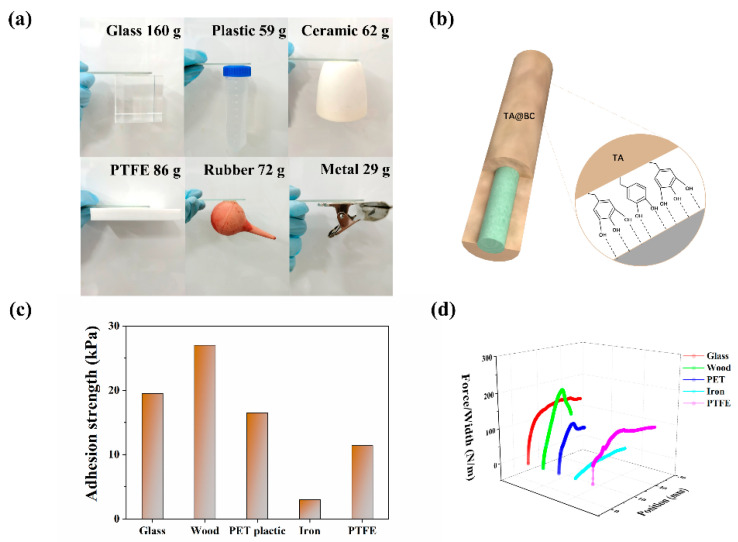
Adhesion properties of the hydrogel. (**a**) Optical images of P(AA-AMPS)-TA@BC_0.5_-Ca^2+^ hydrogel adhesion to different substrate materials; (**b**) schematic illustrations of the adhesion mechanism; (**c**) tensile adhesion strength of P(AA-AMPS)-TA@BC_0.5_-Ca^2+^ hydrogel to different matrix materials; (**d**) adhesion energy of P(AA-AMPS)-TA@BC_0.5_-Ca^2+^ hydrogel on different matrix materials.

**Figure 6 polymers-14-05316-f006:**
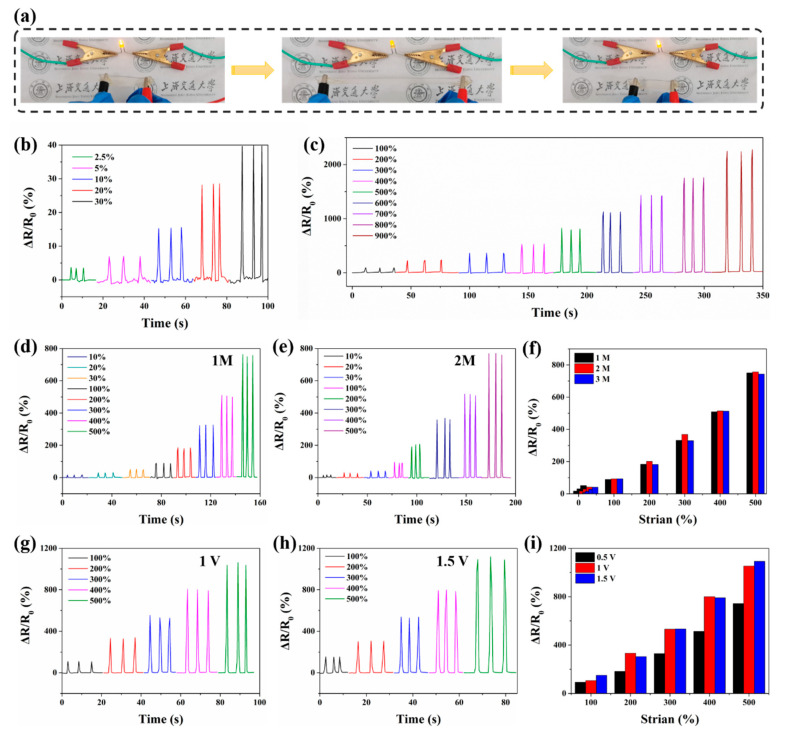
Mechanosensitive performances of the hydrogel. (**a**) The brightness of LED lamp observed at different hydrogel strains; (**b**) the RRCs of hydrogel strip at the elongation form 2.5% to 30%; (**c**) the RRCs of hydrogel strip at the elongation form 100 to 900%; (**d**) mechanical response performance of hydrogel with 1M Ca^2+^ concentration under 10% to 500% strain; (**e**) mechanical response performance of hydrogel with 2M Ca^2+^ concentration under 10% to 500% strain; (**f**) comparison of RRCs values of hydrogels with different Ca^2+^ concentrations; (**g**) under 1 V, mechanical sensing under 10% to 500% strain; (**h**) under 1.5 V, mechanical sensing under 10% to 500% strain; (**i**) comparison of RRC values of hydrogels under diverse voltage.

**Figure 7 polymers-14-05316-f007:**
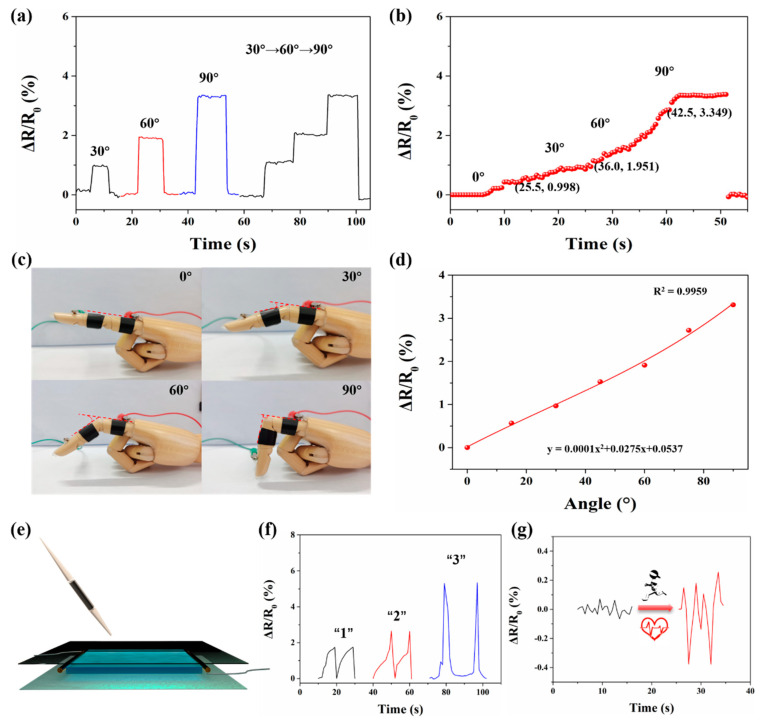
(**a**) RRCs from finger motion of different angles (0°, 30°, 60°, 90°). (**b**) RRC values change under continuous and slow bending; (**c**) optical picture corresponding to finger bending; (**d**) corresponding curve of bending angle and RRC value; (**e**) schematic illustration of the setup for handwriting sensing; (**f**) RRCs for “1”, “2”, and “3” written on the PET film; (**g**) RRCs of the pulse beating before and after running for 5 min.

## Data Availability

Data is contained within the article and Appendix A.

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
