# Peer review of "High Multi-Environmental Mechanical Stability and Adhesive Transparent Ionic Conductive Hydrogels Used as Smart Wearable Devices"

_polymers, 2022, doi:10.3390/polym14235316_

Round 1

Reviewer 1 Report

This well-written manuscript on multifunctional conductive hydrogels investigates their mechanical properties, ionic conductivity at low temperatures, and performance as a function of ion concentration, and voltage. A few questions/suggestions to the authors are the following:

1-      The introduction section provides a good number of references and description of the many challenges associated with the fabrication, design and practical requirements in developing smart wearable sensor devices; however, it is less clear how this work differs from previous work published by the same authors. For instance, it is not clear if this is the first time that this hydrogel was prepared.

2-      A suggestion to the authors is to include FT-IR data in the manuscript, instead of being shown in the supporting information.

3-      Since the hydrogels loose about 70% of their weight after 120h under natural environmental conditions, could the authors comment on how the ionic conductivity and adhesion properties will change for these dehydrated hydrogels?  Is this effect reversible, meaning could the hydrogels be “moisturized” again and recover these properties?

4-      Are the sensors susceptible to any degradation by oxidation?
